# Automated Detection for Concrete Surface Cracks Based on Deeplabv3+ BDF

**Yonggang Shen** [1,2]**, Zhenwei Yu** [1,3]**, Chunsheng Li** [4]**, Chao Zhao** [4] **and Zhilin Sun** [1,]*****

1   College of Civil Engineering and Architecture, Zhejiang University, Hangzhou 310058, China
2   Center for Balance Architecture, Zhejiang University, Hangzhou 310028, China
3   The Architectural Design & Research Institute of Zhejiang University Co., Ltd., Hangzhou 310027, China
4   Zhejiang Communications Construction Group Co., Ltd., Hangzhou 310051, China
*   Correspondence: oceansun@zju.edu.cn; Tel.: +86-10-13606705667

**Abstract:** Concrete cracks have always been the focus of research because of the serious damage they cause to structures. With the updating of hardware and algorithms, the detection of concrete structure surface cracks based on computer vision has received extensive attention. This paper proposes an improved algorithm based on the open-source model Deeplabv3+ and names it Deeplabv3+ BDF according to the optimization strategy used. Deeplabv3+ BDF first replaces the original backbone Xception with MobileNetv2 and further replaces all standard convolutions with depthwise separable convolutions (DSC) to achieve a light weight. The feature map of a shallow convolution layer is additionally fused to improve the detail segmentation effect. A new strategy is proposed, which is different from the two-stage training. The model training is carried out in the order of transfer learning, coarse-annotation training and fine-annotation training. The comparative test results show that Deeplabv3+ BDF showed good performance in the validation set and achieved the highest mIoU and detection efficiency, reaching real-time and accurate detection.

**Keywords:** damage detection; non-destructive evaluation; deep learning; concrete structure; crack segmentation

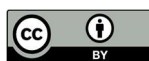

## 1. Introduction

In recent years, many concrete infrastructures have suffered from structural degradation due to long-term, high-load operation, or are close to the end of their natural service life, resulting in safety problems. Therefore, it is necessary to regularly inspect the health status of infrastructure, and the identification and evaluation of structural surface cracks are the tasks that managers and researchers are focusing on. However, traditional manual crack detection methods are inefficient and subjective. How to develop and promote more effective and reliable detection methods is the current research direction.

In view of the limitations of artificial crack detection, Yeum et al. [1] have carried out a lot of research on intelligent crack identification methods in the last decade. Initially, image processing techniques (IPTs) were used to carry out detection tasks, but this method requires additional pre-processing and post-processing technologies, thus, reducing the degree of intelligence. Deep learning algorithms [2], which can automatically extract the sensitive features of the target in the training process, were subsequently proposed and widely studied. Among them, the most representative algorithms are of two types: objection detection [3] and semantic segmentation [4]. Objection detection gives the category and position of the target in an image in the form of a rectangular box, and some models with excellent performance have been proposed, such as the YOLO series and SSD [5] one-stage models and Fast R-CNN [6] and Faster R-CNN [7] two-stage models. Improved models have also been put forward according to specific task requirements. Park et al. [8] proposed a structural crack detection and quantification method in which

YOLOv3-tiny is used to locate concrete cracks in real time. Zhao et al. [9] proposed a crack feature pyramid network (Crack-FPN), which has superior feature extraction capability and reduced computational cost. Some research or reviews on the application of objection detection algorithms to crack detection have also been carried out [10–16]. However, due to the simplicity of the result form, the target detection algorithm is only applicable to simple target existence determination.

Crack images and datasets are highly class imbalanced, and cracks usually have complex textures. According to these characteristics, the research has further developed from objection detection to semantic segmentation algorithm, that is, all pixels belonging to the same type of target are represented by a monochrome mask, and the picture is simplified into a combination of multiple different color masks. Some research has also been carried out on the application of the semantic segmentation algorithm in crack detection [17–22]. Xiang et al. [23] proposed a crack segmentation method based on super-resolution reconstruction, which achieved a more than 10% performance improvement compared with previous models but could not meet the real-time requirements. Ren et al. [24] proposed a new end-to-end crack segmentation method based on a fully convolutional network which uses dilated convolution, spatial pyramid pooling, skip connection and an optimization loss function to obtain higher efficiency and accuracy. However, researchers have ignored or avoided some aspects of research, such as: (1) The computation amount required by the semantic segmentation algorithm is very large, and it even takes a few seconds to detect an image in the early stage. If the semantic segmentation is intended to be used in an actual scene, the data type is usually video with a frame rate of 60. When considering frame extraction or reducing the frame rate, real-time detection requires that the model processes images at a speed of 0.033 to 0.04 s per image, that is, 25 to 30 frames per second (FPS). At present, the detection efficiency of many models is difficult to achieve in actual projects. (2) Due to the limitations of manual labeling, the division between the cracks and the background boundary in the label is relatively vague, which makes the segmentation results given by the trained model show a large number of false positives and false negatives on the boundary [25]. (3) Mei et al.'s study [26] and many other studies deployed a transfer learning [27] strategy in a model, using initial weights trained on a large dataset containing many categories. These datasets have many objects of different classes from cracks (for example, ImageNet [28] has more than 5000 classes), and the extracted features are not highly related to cracks. Even if the models are continuously trained with carefully prepared datasets after transfer learning, it usually takes a lot of time to complete the production of segmentation labels. It seems to be a method to automatically label targets with computers, but the marking model still requires an initial dataset to complete training before it can be put into use.

In this paper, a new pixel-level semantic segmentation model for crack detection based on Deeplabv3+ [29] is proposed to solve the above problems and is named Deeplabv3+ BDF. This model can overcome the interference of background and crack-like features, extract the crack boundary quickly and accurately and, thereby, prepare for the intelligent detection of fine indicators such as crack width across complex background, so that the management and maintenance department can concentrate resources to study cracks and ignore the background or other objects. In addition, this paper also attempts to use a new training strategy to reduce the common labeling cost problem of semantic segmentation models, which provides a potential solution for researchers with a large amount of data but not enough resources to fine-label all data. The main contributions of this paper are as follows:

(1) A lightweight network MobileNetv2 is used as the backbone, and all standard convolutions are replaced by DSC to reduce the number of parameters and realize real-time detection;

(2) On the basis of the characteristics of semantic hierarchy and cracks, during the up-sampling process, the shallow feature map after one down-sampling is fused to

improve the segmentation accuracy at the boundary between the foreground and background;

(3) Focusing on the problem of the labeling cost of semantic segmentation model being too high, a three-step training strategy according to the sequence of transfer learning, coarse-annotation (CA) training and fine-annotation (FA) training is designed and proposed, which can enhance the learning and extraction of crack features. This training strategy can train a better segmentation model with a large number of CA images on the premise of only a few FA images, saving a lot of human and material resources.

## 2. Models and Methodology

### 2.1. Deeplabv3+

The Deeplab series was developed on the basis of FCN [30]. Its main feature is to expand the receptive field by using atrous spatial pyramid pooling (ASPP) to obtain more image feature information. Deeplabv3+ achieves 87.8 mIoU on the PASCAL VOC-2012 dataset, and its image segmentation effect is superior to other Deeplab series models. Compared with Deeplabv3, the main feature of Deeplabv3+ is that it adds a decoder module with transposed convolution as the main unit, which can gradually restore high-dimensional feature vectors to the feature map of the same size as the input image. Figure 1 shows the network diagram of Deeplabv3+.

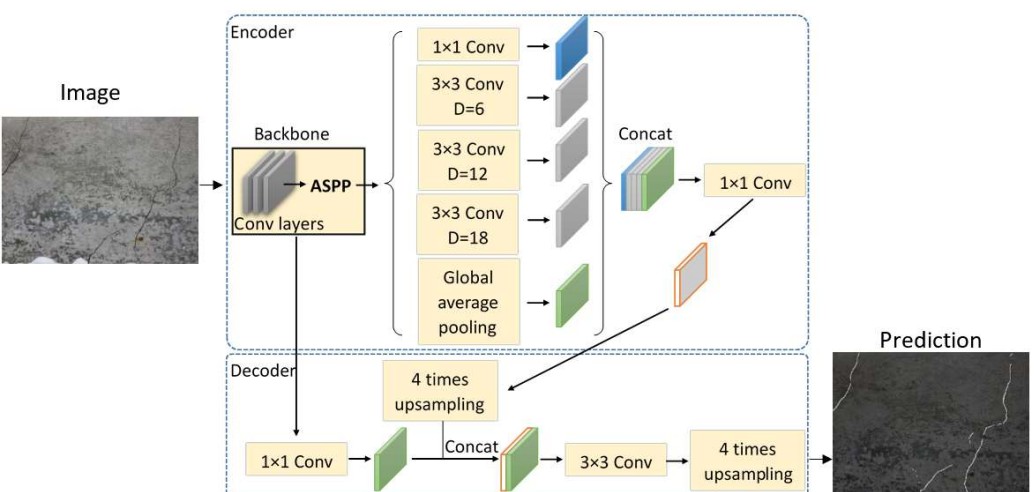

**Figure 1.** The structure of Deeplabv3+.

The encoder consists of backbone network Xception [31] and ASPP. Xception extracts two feature maps of high semantic information and low semantic information at the same time. The former usually represents an abstract concept and is the information expressed by the image closest to human understanding, while the latter is the color, texture and shape. The high semantic information feature map conducts multi-scale, dilated convolution sampling in the ASPP module, generates and fuses multiple feature maps of different scales and, finally, uses 1 × 1 convolution for dimension reduction. Low semantic information is transferred into the decoder part for 1 × 1 convolution and is fused with the high semantic information feature map after bilinear up-sampling four times to enhance the network learning effect and improve the segmentation accuracy. Then, the feature is extracted through 3 × 3 convolution, and the final semantic segmentation map is obtained by up-sampling four times.

### 2.2. Deeplabv3+ BDF and Optimization Strategies

According to some defects in the current research described in Section 1, including defects relating to detection efficiency, boundary ambiguity and initial weight mismatch, we propose a method to improve the model accordingly and rename the model according to the optimization strategy used, namely Deeplabv3+ BDF.

### 2.2.1. Backbone

MobileNetv2 is used to replace Xception as the backbone. MobileNetv2 is the same as MobileNetv1, which is a lightweight CNN, and it uses DSC. By adjusting the number of channels in each convolution layer, MobileNetv2 does not affect the performance but can reduce the amount of computation. Taking it as a backbone can effectively improve the detection speed and decrease the occupation, making the model oriented to real-time detection. For more details, please refer to [32].

### 2.2.2. DSC

All standard convolutions other than the backbone are replaced by DSC [33], including standard convolutions in the decoder to accelerate the detection. DSC can be divided into depthwise convolution (DWC) and pointwise convolution (PWC) [34]. The comparison of these convolution operations is shown in Figure 2. Take the convolution operation in Figure 2 as an example. There are four filters in the standard convolution, and each filter has three convolution kernels, which correspond to three channels of the image. After convolution, the feature maps with the same number of filters is obtained, and the parameter quantity is $4 \times 3 \times 3 \times 3 = 108$. There is only one convolution kernel in each filter of DWC, which is responsible for one channel, respectively. The number of channels before and after convolution remains unchanged, and the parameter quantity is $3 \times 3 \times 3 = 27$. The convolution kernel size of PWC is $1 \times 1$, and its function is to generate a new feature map by weighted combination of the output feature maps of the upper layer. It is a special case of standard convolution when convolution kernel size is $1 \times 1$ and the parameter quantity is $1 \times 1 \times 3 \times 4 = 12$. After DWC and PWC, a four-channel output can also be obtained, which is the same as in standard convolution. Moreover, compared with the standard convolution, the parameter quantity of DSC is $27 + 12 = 39$, which is only 36.1% of the standard convolution, and the calculation cost is significantly reduced.

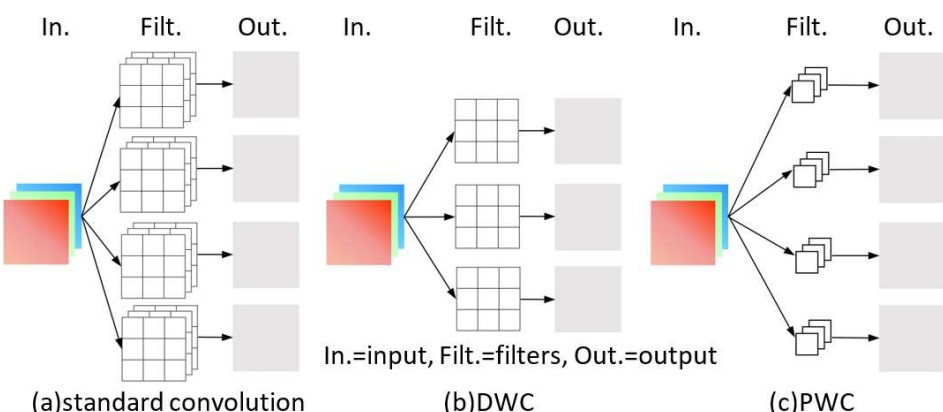

**Figure 2.** Comparison of convolution operations.

### 2.2.3. Feature Fusion

The feature map after one down-sampling is fused additionally, as shown in Figure 3. Convolution features are hierarchical and choosing different layers may achieve completely different results. Shallow features of CNN focus on detail features, such as edges and corners, which are usually associated with individual segmentation results. Middle features are a part of the object, and deep features focus on deeper semantic information.

Intuitively, deep feature maps can represent a complete object, which is usually related to the accuracy of classification results. Only when the receptive field size of the feature map is larger than the object can the correct detection be carried out. Correspondingly, shallow features can only cover small objects, while deep features can cope with larger objects.

The boundary of the crack belongs to edge or corner features, so the feature map after one down-sampling is added to the up-sampling process to enhance the ability of the model to deal with the demarcation. For the feature map after three down-samplings, because the crack itself is a tiny object, even if it runs through the whole image, the number of pixels belonging to the crack is very small. The cracks contained in the receptive field corresponding to the feature map after two down-samplings can meet the training requirements, because a part of a long crack can still be regarded as a crack with complete features. However, it is also unreasonable to only fuse the feature map after one down-sampling, because it needs a larger field of vision to judge whether it is a crack or a crack-like object. Only focusing on the edges or corners of the object may mistake some black, slender objects for cracks, thus, we retain the strategy of fusing the feature map after two down-samplings in Deeplabv3+.

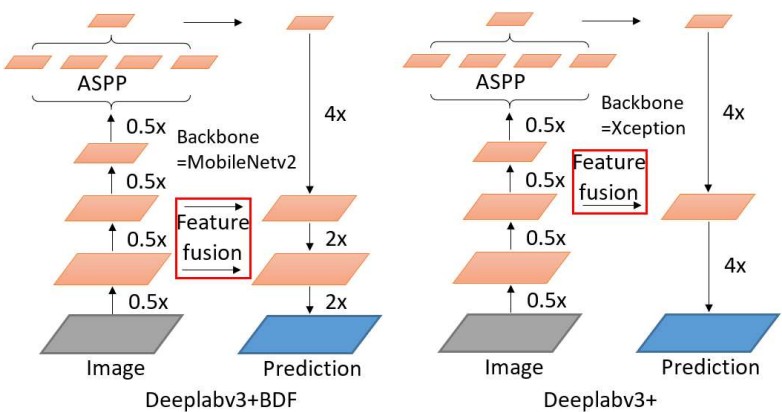

**Figure 3.** Comparison of network structure changes.

## 3. Establishment of Dataset

The biggest difference between the actual project and the laboratory scene is the environment around the crack. Generally, images obtained in the field experience interference due to handwriting, template lines and other crack-like objects, while the crack images obtained in the laboratory have a monotonous background and no sundries; so, the trained model is difficult to extend to practical applications. Therefore, we take 82 images from multiple scenes. The illumination conditions, exposure intensity and acquisition equipment of these images are different, so the dataset has enough diversity. Because the dataset is collected by mobile phones or high-definition cameras, the image capture distance varies in a large range (0.2 m to 5 m), and the image scene also has enough complexity; the trained model has good recognition effect on common crack images. However, the model has the potential to improve the recognition effect of fine cracks, especially for cracks where the width is less than 1 pixel, which are very easy to be missed in detection. This is also the difficulty of the semantic segmentation model used for crack detection at present. Examples of the dataset are shown in Figure 4. Four images at 3840 × 2880 pixels are from a composite plate failure experiment, eight images at 1920 × 1080 pixels are from another composite plate experiment, four images at 4608 × 3456 pixels are from a bridge, seven images at 4608 × 3456 pixels are from some cracked walls or structures in Zhejiang University, five images at 1920 × 1080 pixels are from a concrete beam bending experiment and five images at 1920 × 1080 pixels are from a concrete column bending experiment. These images are manually labeled at pixel level using the Labelme program. Another 49 images at 1920 × 1080 pixels are obtained from a concrete beam bending test, and CAs are made to enable the model to be pretrained. Although transfer learning is an effective

strategy, its initial weights are usually trained by multi-class objects, and there are a lot of irrelevant or weak correlation features. After transfer learning, more pretraining for cracks can weaken these irrelevant or weak correlation features and strengthen the recognition and extraction of crack features.

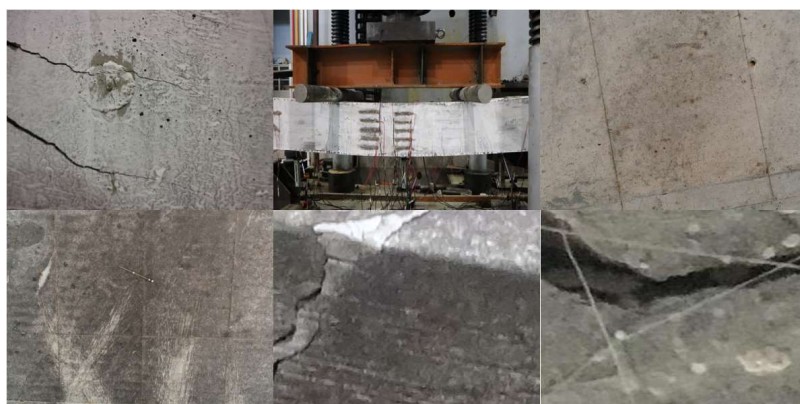

**Figure 4.** Examples of the dataset.

The comparison of CA and FA is shown in Figure 5. CAs use more long lines and obvious angles to represent the irregular contour of the crack. The form shown in the figure shows that the mosaic area is wider, the jagged boundary of the crack is ignored and the final label can be approximately regarded as a polygon, which greatly reduces the time for CA. FAs make the marking points fit the crack as much as possible so that the width of the mosaic and the outline of the label basically match the crack. Therefore, obvious bending does not appear in the figure, and the overall appearance of FA is smoother. It takes about 6 min for an image to be coarsely labeled and 15 min to be finely labeled. This method of training based on two types of labels does not require fine labeling of all images and can reduce labeling costs.

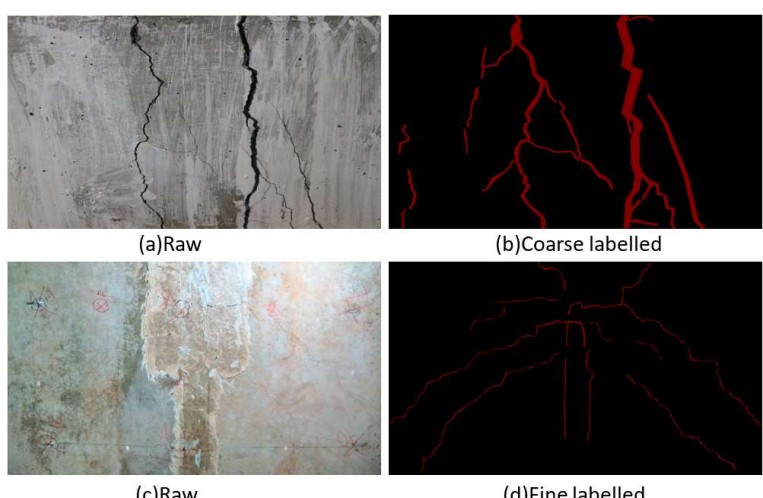

**Figure 5.** Comparison of CA and FA.

Firstly, all images are divided into sub-images at 576 × 576 pixels; 588 coarse-labeled images and 944 fine-labeled images are obtained. Then, images without cracks are removed, and 550 coarse-labeled images and 676 fine-labeled images are obtained. Due to the small amount of data, in order to make the test results more reliable, we refer to a data configuration strategy similar to K-fold verification in [35] so as to avoid over-fitting. The dataset is divided into five subsets for cross-training and validation. The division of sub-datasets is shown in Table 1, and the dataset not used in each training process is used as

the code of this training process; so, each sub-dataset includes 540/541 images (including only FA images) or 980/981 images. CA images are not used for testing.

Due to the small amount of data, the model proposed in this paper should be limited to detecting concrete cracks, and the prediction effect of surface cracks in extreme environments (such as earthquake) or other materials needs further research.

**Table 1.** Division of sub-datasets.

| Sub-Dataset No. or Training Process | CAs | FAs |
|---|---|---|
| Sd1 | 110 | 135 |
| Sd2 | 110 | 135 |
| Sd3 | 110 | 135 |
| Sd4 | 110 | 135 |
| Sd5 | 110 | 136 |
| Total | 550 | 676 |

## 4. Model Training and Results

The contents of this chapter include the details and experimental results of training Deeplabv3+ BDF. The optimization of the model is implemented using Python and the open-source framework Deeplabv3+. The computing workstation is configured with four 1080 Ti GPUs. In the following paper, transfer learning is referred to as process T, training on CA images is referred to as process C and training on FA images is referred to as process F.

### 4.1. Training Strategy and Experimental Results

Training is divided into four types: F, C + F, T + F and T + C + F, i.e., three-step training. Process T does not need to be specific, and using the Cityscapes initial weights already available in the Deeplabv3+ model package can be considered an alternative to this process. Deeplabv3+ BDF is trained on CA images firstly and then on FA images as the model converges. Experiments show that process C converges after 15,000 epochs, and the weight of any subsequent epoch can be used as the initial weight of process F. In this paper, process C is iterated with 20,000 epochs. The image input size is set to 577 × 577 resolution, the loss function is binary cross-entropy loss, the initial learning rate is 0.0001, the learning rate attenuation coefficient is 0.1, the number of attenuation steps is 2000, the batch size is 32, the dropout rate is set to 0.5 and the total number of epochs is 50,000. That is, when process C with 20,000 epochs exists, process F continues to iterate 30,000 epochs, and the loss is recorded every 10 epochs.

In order to present the figures clearly, a simple moving average (SMA) of every 500 steps is used to describe the loss curve, as shown in Figure 6. The SMA is calculated according to Equation (1):

$$\text{SMA} = \begin{cases} (L_i + L_{i-1} + L_{i-2} + \cdots + L_1)/i, & if\ i \in [1,500) \\ (L_i + L_{i-1} + L_{i-2} + \cdots + L_{i-499})/500, & if\ i \in [500, 50000] \end{cases} \quad (1)$$

where $i$ is the number of iterations, and $L_i$ is the loss value of the $i$th iteration. Since the number of iterations is less than 500, SMA calculation needs to follow another variant form in the first 499 iterations, while SMA is normally calculated after 500 iterations. After two pretraining sessions of process T and process C, the three-step training model still shows the potential to be optimized in process F and further decreases in the loss value and finally converges to 0.25, which is the lowest of the four training strategies. Even for process C + F without process T this phenomenon also appears, that is, after the process C training weight excellence, it still has room to be improved, and its loss converges to 0.39.

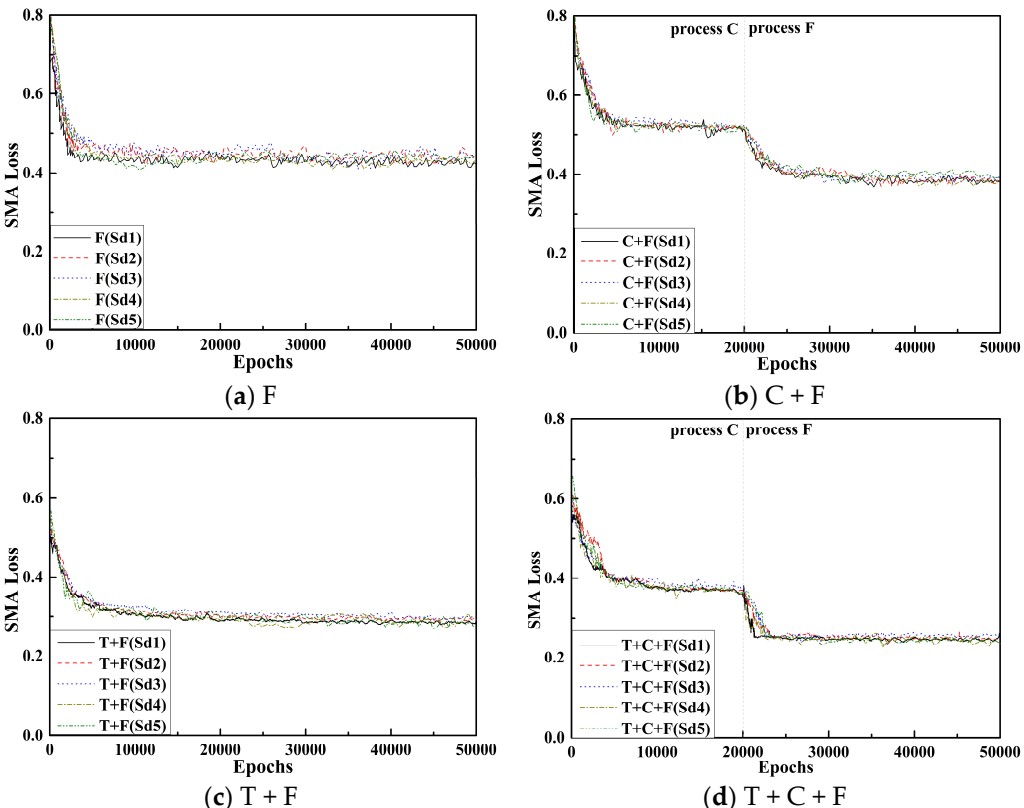

**Figure 6.** SMA loss curve.

Process T + F and process F, which are two conventionally used training strategies with convergence values of about 0.30 and 0.42, respectively, without process C can also converge but are higher than T + C + F. In the two strategies without process T, the SMA curve fluctuates in different degrees. Note that this is the average curve, and its fluctuation is somewhat mitigated, but it still has disadvantages compared with the corresponding curves of the two strategies with process T. Therefore, process T is necessary.

The effectiveness of our proposed training strategy can also obtain the same conclusion from the validation indices. The precision (P), recall (R), F1-score and the most commonly used index in semantic segmentation task, mIoU, are used for evaluation, and they are calculated according to Equations (2)–(5):

$$P = TP/(TP + FP), \tag{2}$$

$$R = TP/(TP + FN), \tag{3}$$

$$F1\text{-score} = 2 \times P \times R/(P + R), \tag{4}$$

$$mIoU = (1/k) \times TP/(FN + FP + TP), \tag{5}$$

where TP, FP and FN represent true positive, false positive and false negative, respectively, and K is the number of target categories in all images. Figure 7 shows that, when the training strategy is T + C + F and the dataset is Sd2, the change of mIoU curve is opposite to the loss curve, but the change trend is the same, that is, when process F is carried out after the convergence of process C, the curve has a certain mutation. The mIoU curves of the four training types on the Sd1 subset are shown in Figure 8. From the comparison between F and C + F, and the comparison between T + F and T + C + F, it can be seen that, although the curves of C + F and T + C + F are lower at the initial stage, the inversion is

achieved after 20,000 iterations, which indicates that the models with CA and FA training have significant optimization in terms of mIoU.

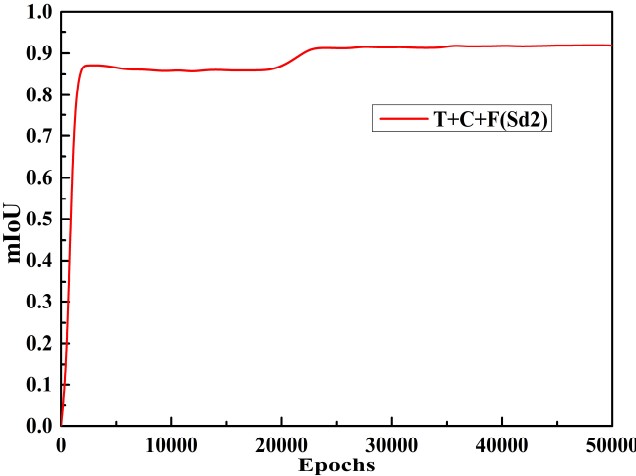

**Figure 7.** Example of mIoU curve.

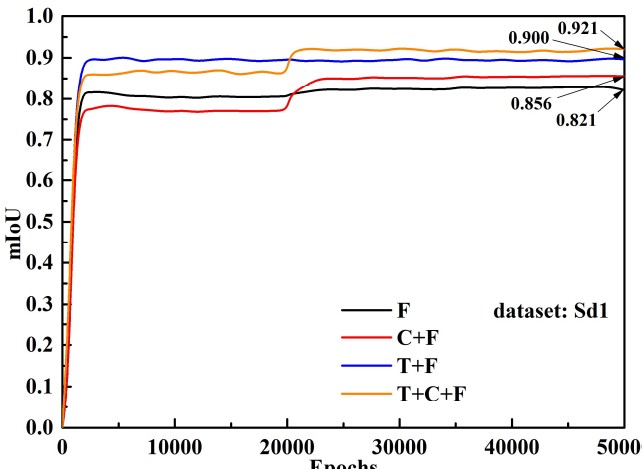

**Figure 8.** mIoU curves of four training types.

The performance of the four training strategies on the validation set is shown in Table 2, which shows the indices of different training strategies on different sub-datasets. The results of process T + C + F is the highest in the four indices. From the comparison of T + F and F, T + C + F and C + F, it is revealed that transfer learning is still a very effective strategy when dealing with small datasets. It can provide better initial performance, optimization rate and convergence for the model. The effectiveness of establishing a training strategy for a specific task can be proved by the comparison of T + F and T + C + F and the comparison of F and C + F. Deeplabv3+ BDF that is pretrained with CA data for a specific task and then trained normally is better at indicators than the model that is trained directly on FA data. The conventional training strategy, i.e., process T + F, is 0.019 lower than the secondary pretraining method proposed in this paper in terms of mIoU.

**Table 2.** Indicators obtained by different training strategies.

| Training Strategy | Training Process | P | R | F1 | mIoU |
|---|---|---|---|---|---|
| F | Sd1 | 0.837 | 0.841 | 0.839 | 0.821 |
| | Sd2 | 0.843 | 0.838 | 0.840 | 0.824 |
| | Sd3 | 0.831 | 0.828 | 0.829 | 0.816 |
| | Sd4 | 0.834 | 0.829 | 0.831 | 0.818 |
| | Sd5 | 0.839 | 0.835 | 0.837 | 0.820 |
| | Average | 0.837 | 0.834 | 0.835 | 0.820 |
| C + F | Sd1 | 0.871 | 0.874 | 0.872 | 0.856 |
| | Sd2 | 0.863 | 0.859 | 0.861 | 0.841 |
| | Sd3 | 0.854 | 0.865 | 0.859 | 0.838 |
| | Sd4 | 0.861 | 0.868 | 0.864 | 0.845 |
| | Sd5 | 0.859 | 0.851 | 0.855 | 0.843 |
| | Average | 0.862 | 0.863 | 0.862 | 0.845 |
| T + F | Sd1 | 0.883 | 0.880 | 0.881 | 0.900 |
| | Sd2 | 0.875 | 0.874 | 0.875 | 0.893 |
| | Sd3 | 0.881 | 0.877 | 0.879 | 0.902 |
| | Sd4 | 0.879 | 0.877 | 0.878 | 0.899 |
| | Sd5 | 0.874 | 0.879 | 0.876 | 0.895 |
| | Average | 0.881 | 0.877 | 0.879 | 0.898 |
| T + C + F | Sd1 | 0.892 | 0.903 | 0.897 | 0.921 |
| | Sd2 | 0.889 | 0.897 | 0.893 | 0.917 |
| | Sd3 | 0.883 | 0.887 | 0.885 | 0.912 |
| | Sd4 | 0.887 | 0.882 | 0.884 | 0.914 |
| | Sd5 | 0.892 | 0.896 | 0.894 | 0.919 |
| | Average | 0.889 | 0.893 | 0.891 | 0.917 |

Some experiments are carried out on the proportion of data required by process C and process F, and the final mIoU is taken as the evaluation index. A total of 550 CA images are divided into five sub-datasets. Trial training is conducted according to the number of sub-datasets from 1 to 5, and the mIoU of each experiment is recorded, which can be seen in Figure 9. The results show that the difference of mIoU is within ±0.03 after using three or more sub-datasets, i.e., 330 CA images, which can be regarded as the fully developed optimization potential. However, this result is only for the task of this paper. In an actual project, the number ratio of CA and FA images should be determined according to the complexity of the task, the characteristics of the object and other factors. Section 3 describes the time spent in annotation, and an FA image is 2.5 times a CA image. If 330 or more CA images are labeled according to the FA image standard, it takes more time to achieve the same result, and this problem can be solved using the three-step training strategy proposed in this paper.

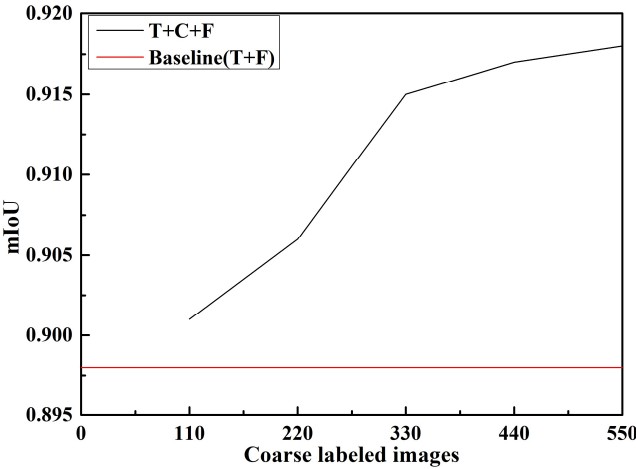

**Figure 9.** The change of mIoU with the number of CA images.

*4.2. Comparison Results*

The excellent performance of Deeplabv3+ BDF can also be shown in the comparison experiment, and we compare Deeplabv3+ BDF with a variety of representative semantic segmentation models, including Deeplabv3+, U-Net [36], PSP-Net [37], DeepCrack [38] and DeepCrack-Aug [38], among which DeepCrack is a model specially designed for crack detection. Table 3 shows the results of the comparative experiment. Due to the use of a lighter backbone and the replacement of all standard convolutions with DSCs, the number of parameters in Deeplabv3+ BDF is greatly reduced, the physical occupation and running video memory occupation are reduced and the detection speed is greatly improved. In addition, even Deeplabv3+ BDF without process C intensive training is superior to other models in various evaluation indicators, while Deeplabv3+ BDF with process C training further expands its advantages. The mIoU of Deeplabv3+ BDF (T + C + F) takes the lead over Deeplabv3+ with better performance at 0.102, and the detection speed reaches 26.132FPS, 2.9 times faster than U-Net. The comparison experiment proves that our optimization measures are effective. Deeplabv3+ BDF has both accuracy and speed and can meet real-time detection requirements (generally 20–25 FPS). If Deepalabv3 + BDF (T + C + F), Deeplabv3+ and DeepCrack-Aug are used simultaneously to detect 100 images at 576 × 576 pixels, which takes 3.83 s, 26.56 s and 11.43 s, respectively, our model can complete the same task with high accuracy and speed, and this advantage is more obvious when the number of images is larger.

**Table 3.** Comparison results using the dataset of this paper.

| Model | Deeplabv3+ | U-Net | PSP-Net | DeepCrack | DeepCrack-Aug | Deeplabv3+ BDF (T + F) | Deeplabv3+ BDF (T + C + F) |
|---|---|---|---|---|---|---|---|
| P | 0.821 | 0.813 | 0.801 | 0.467 | 0.512 | 0.880 | 0.888 |
| R | 0.783 | 0.779 | 0.724 | 0.532 | 0.502 | 0.877 | 0.897 |
| F1 | 0.802 | 0.796 | 0.761 | 0.497 | 0.507 | 0.878 | 0.892 |
| mIoU | 0.815 | 0.782 | 0.746 | 0.612 | 0.538 | 0.898 | 0.917 |
| FPS | 3.765 | 8.934 | 7.824 | 2.784 | 8.752 | 25.783 | 26.132 |

The results of crack semantic segmentation detection can be used to measure the width and length of cracks, and its application in the field of structural damage detection is not only as demonstrated here, but is also for corrosion detection and calculation and statistics of corrosion area, which will be carried out in our future work.

### 4.3. Typical Inference Results

The proposed three-step training method is proved to be effective. In this section, we use representative inference results to show that Deeplabv3+ BDF with additional feature map fusion has better performance. Another training session is conducted using the same strategy and dataset Sd1, but the training object is changed to Deeplabv3+ BDF, which only fuses the feature map twice, and it is named Deeplabv3+ BDF-single for differentiation. These two models are inferred from the same test set, three representative images are selected and the mIoU values after detection are attached. The pixels in the segmentation result can be divided into four categories, namely, TP in red, FN in blue, FP in green and TN representing the background, as shown in Figure 10.

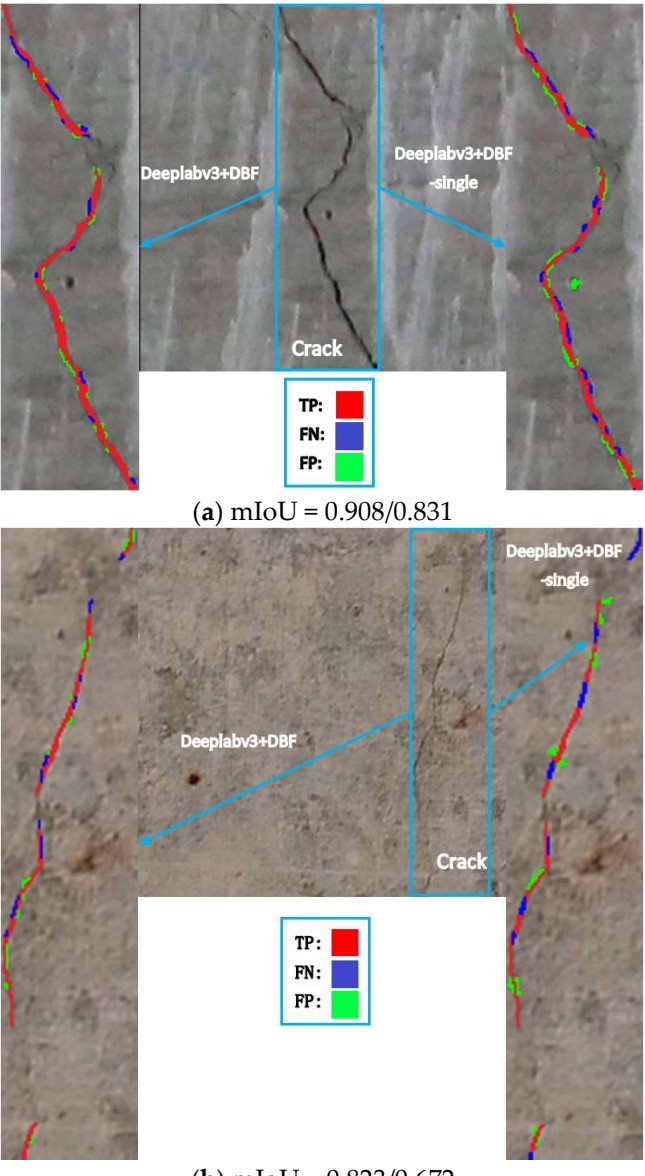

(**a**) mIoU = 0.908/0.831

(**b**) mIoU = 0.823/0.672

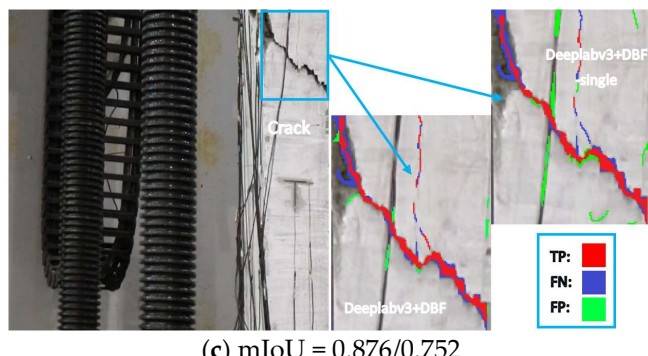

(**c**) mIoU = 0.876/0.752

**Figure 10.** Typical segmentation results.

Deeplabv3+ BDF gives more accurate segmentation results of all fractures and obtains significant advantages over the comparison model in mIoU. Figure 10a is a crack image containing a small cavity with simple background. Deeplabv3+ BDF-single incorrectly identifies the cavity as a crack, but Deeplabv3+ BDF does not. The segmentation results show that the mIoU of Deeplabv3+ BDF-single is only 0.831 even in simple background, and more FP pixels appear on the dark concrete background at the crack edge. Figure 10b contains a fuzzy fine crack. Fine cracks are the key and difficult task of crack detection. However, due to the accuracy of human eyes, the edges of cracks cannot be accurately marked, which makes it difficult to completely segment the fine cracks. Deeplabv3+ BDF has less truncation on the whole fine crack and less error expansion at the crack edge, while Deeplabv3+ BDF-single detects a complete fine crack as nearly 10 cracks, and the mIoU also reflects the performance difference from the data level. Figure 10c contains a spalling concrete surface, and the crack that passes through it can be confusing. Deeplabv3+ BDF overcomes the problem of crack area expansion caused by spalling to a certain extent, but there are a lot of FN results in the spalling area, which still has the potential for improvement. Deeplabv3+ BDF-single has a similar problem in dealing with the problem of spalling concrete. The whole crack is segmented more thinly, and the detection of the next thin cracks still fails, and the instrument circuit is classified as crack, which is unacceptable. It can be seen from Figure 10 that most of the FP and FN values of the crack segmentation task appear at crack boundary, which is the reason why we choose to fuse the shallow feature map, because it corresponds to the edges and corners of crack. Although the judgment of crack boundary is subjective due to the different degrees of image blur, at an overall level, however, the additional feature map fusion strategy reduces the proportion of FP and FN and has advantages in filtering various complex backgrounds or processing cracks with different widths.

Figure 11 shows the segmentation results of our model, Deeplabv3+, U-Net and DeepCrack-Aug. Figure 11a shows a crack with an average width of about 3.7 pixels. Our model has almost the same result as the groundtruth. The worst one is DeepCrack-Aug. As mentioned earlier, the dataset used by DeepCrack is relatively simple, and it is easy to make mistakes in some common images with general complexity. Figure 11b shows a crack with an average width of only 1.6 pixels. In this image, all models have made many errors, but, from a comprehensive perspective, our model still achieves the best results. The other three models have different degrees of truncation and error expansion for this fine crack. It can also be inferred from Figure 11b that our model can identify cracks only 1 pixel wide and separate them from the background. It is proved that our model is superior to the recent work in both data and segmentation visualization.

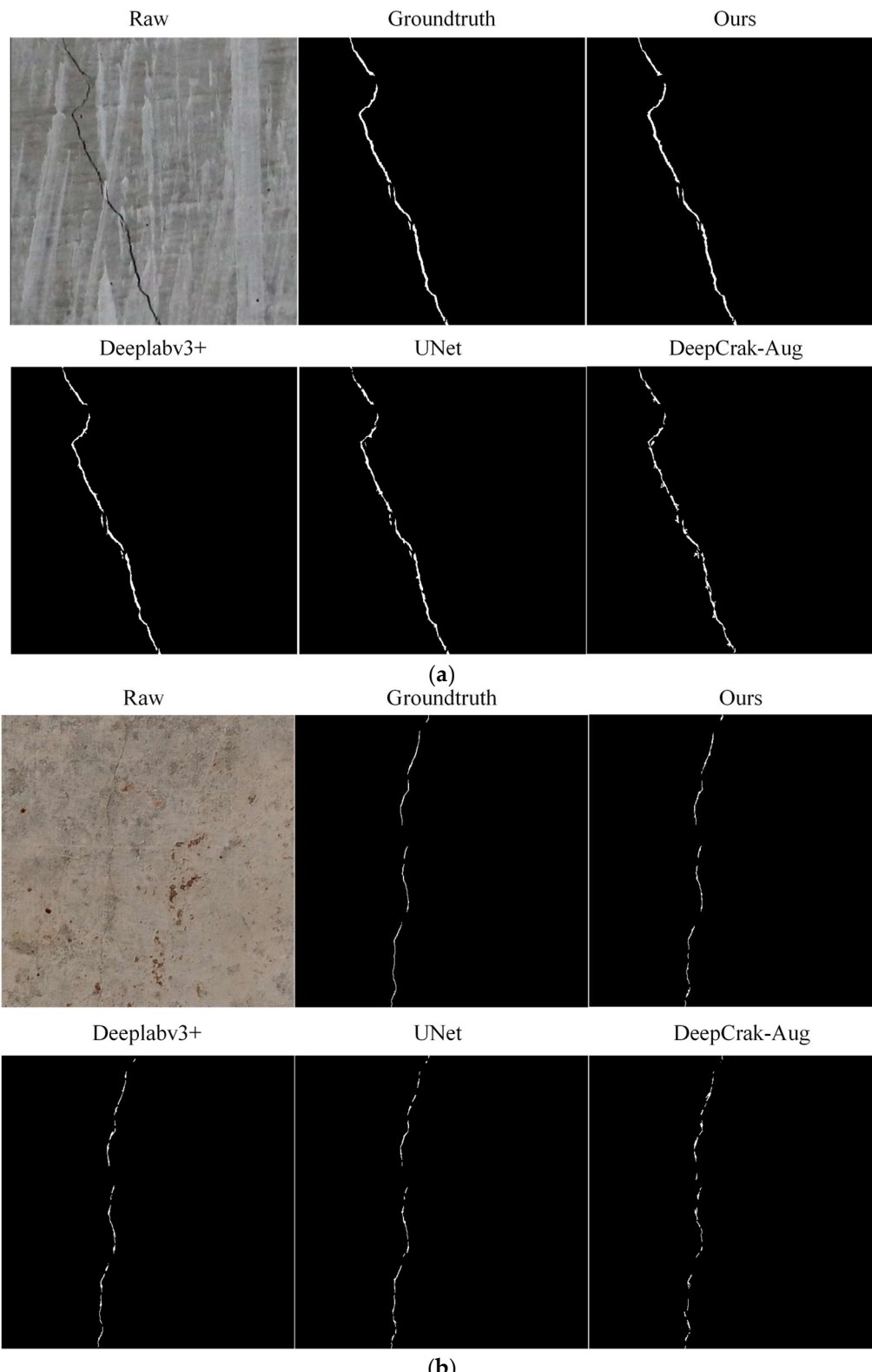

**Figure 11.** Segmentation results comparison of multiple models. Subfigure (**a**,**b**) are two randomly selected crack images.

## 5. Conclusions

In this study, we noticed that the current research has some defects or ignores some problems, so we proposed a semantic segmentation model improving on Deeplabv3+ and named the model Deeplabv3+ BDF according to optimization strategies. The identification

of cracks is the most critical task in structural health detection, so we think it is necessary and beneficial to propose a deep learning network dedicated to crack detection according to the unique characteristics of cracks. We adjusted the training strategy of Deeplabv3+ BDF and optimized the network structure and established a dataset including CA images and FA images. After the training, we evaluated the performance of the proposed Deeplabv3+ BDF by comparing it with other models. The results show that Deeplabv3+ BDF solves the three problems mentioned in the paper well, especially realizing real-time detection. The conclusions are as follows:

(1) The network structure of Deeplabv3+ BDF is made lightweight by using MobileNetv2 as the backbone network, so that its FPS of 576 × 576 pixels image is 26.132, which meets the real-time requirements and is 2.9 times faster than in recent works;

(2) Because of the additional fusion of shallow feature map, Deeplabv3+ BDF can reduce the number of FN and FP values in detection results and improve the processing ability of the boundary between foreground and background under the same conditions;

(3) After the second pretraining, that is, the proposed three-step training strategy, the potential of Deeplabv3+ BDF is further developed. Compared with the conventional training strategy, the mIoU of Deeplabv3+ BDF is increased to 0.917, which is at least 0.102 ahead of other models.

**Author Contributions:** Conceptualization, Y.S., Z.Y. and Z.S.; methodology, Y.S. and Z.Y.; validation, Y.S. and Z.Y.; investigation, Z.Y., C.L. and C.Z.; resources, Y.S. and Z.S.; data curation, Z.Y.; writing—original draft preparation, Y.S. and Z.Y.; visualization, Z.Y.; supervision, Y.S. and Z.S.; project administration, C.L. and C.Z. All authors have read and agreed to the published version of the manuscript.

**Funding:** This research was funded by ZJU-ZCCC Institute of Collaborative Innovation (No.ZDJG2021009).

**Institutional Review Board Statement:** Not applicable.

**Informed Consent Statement:** Not applicable.

**Data Availability Statement:** Not applicable.

**Acknowledgments:** We acknowledge Tutor Huang Yifang at the Training Platform of Construction Engineering at the Polytechnic Institute of Zhejiang University.

**Conflicts of Interest:** The authors declare no conflict of interest.

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
