# Peer review of "Automated Detection for Concrete Surface Cracks Based on Deeplabv3+ BDF"

_buildings, doi:10.3390/buildings13010118_

Round 1

Reviewer 1 Report

The authors presented an open-source software that was optimized through replacing standard convolutions to achieve better image processing for detecting and quantifing surface cracks in concrete structures. Despite the results produced by the updated software exhibit better outcomes compared to the original product, the paper in general provides moderate novelity and insignificant contribution to the existing works. 

Reviewer 2 Report

The study is devoted to the topical and important topic of identification of cracks in concrete structures.

The article is well written, qualitatively designed and the problem that the authors are investigating is relevant.

The article title is clear.

Keywords are chosen correctly and match the direction of the research.

This article contains new aspects, but the authors must underline the major findings of their work, and explain how this study represents a progress comparatively with other papers. Please clearly explain the novelty.

I also recommend adding information about the perspective of practical use of the obtained model.

A list of specific suggestions that could improve the article:

1. The authors use the phrase "the standard of real-time and accurate detection" several times. It is necessary to explain what it is.

2. Lines 39, 41. Please avoid using the expression "Some scholars". Must be replaced by, for example, "Liu et al."

3. Lines 197-199. How is the marking procedure?

4. Line 203. What are "sound images"?

5. Line 211. What software did the authors use?

6. Line 235. It is necessary to describe mathematically and explain the logic of SMA calculation.

7. The article haven't a comparison of the results obtained with similar studies by other scientists. Can the authors estimate the time savings of using DSC or the entire model?

Reviewer 3 Report

The authors have studied the application of Deeplabv3+, a deep learning algorithm, in detecting concrete cracks, which is an interesting study. The following suggestions are provided for the authors to improve this manuscript.

1. line 204: "176 fine labeled images were obtained" does not match the data in Table 1, please check by the authors.

2. the total number of images in the dataset obtained by partitioning 109 images of different resolutions into 576*576 size and the number of each subset of data, please explain in the text.

3. from the representation in the paper, the number of datasets is small, even if migration learning is used for pre-training, is it applicable to concrete crack detection in various scenes captured by different shooting devices? Could the authors please describe the scope of applicability of the model and elaborate on the inapplicable scenarios as appropriate.

4. it is mentioned in the paper that the crack labeling procedure is performed manually, and the large amount of data labeling in practical applications will incur huge time cost, what procedure would the authors use/recommend to manage the large amount of data?

5. could the authors please indicate the range of the model for the identification of concrete cracks (what is the minimum width of cracks that can be identified).

6. could the authors please state in the text what loss function is used in the model?

7. should Table 3 be placed in Section 4.3? Could the authors please check the typography.

8. suggest whether it is more reasonable to switch the order of section 4.2 and section 4.3, with a horizontal comparison of different testing models first, followed by a vertical comparison of different parameters of the same model, to make the manuscript more convincing.

9. there are fewer pictures in the text, and it is suggested that the authors supplement the mIoU graphs of the four training types and the segmentation comparison graphs of different models.

Round 2

Reviewer 1 Report

The authors have addressed the reviewer's comments and made thoughtful revision to the manuscript.